# Content and Bioaccessibility of Vitamin K (Phylloquinone and Menaquinones) in Cheese

**DOI:** 10.3390/foods10122938

**Published:** 2021-11-29

**Authors:** Marie Bagge Jensen, Andrius Daugintis, Jette Jakobsen

**Affiliations:** Research Group for Bioactives—Analysis and Application, National Food Institute, Technical University of Denmark, 2800 Kgs. Lyngby, Denmark; andrius.daugintis@gmail.com (A.D.); jeja@food.dtu.dk (J.J.)

**Keywords:** starter culture, ripening time, in vitro, infogest 2.0, bioavailability

## Abstract

Vitamin K is a fat-soluble group of vitamers consisting of phylloquinone (PK) and menaquinones (MKs). To date, only a daily reference intake for PK is set; however, in the last decade, research studying the correlation between MKs intake and improvement of health in regards to cardiovascular diseases, bone metabolism, and chronic kidney disease has been conducted. MKs are synthesised by bacteria in the fermentation process of foods, e.g., cheeses. The content and bioaccessibility of vitamin K vitamers (PK, MK-4, MK-5, MK-6, MK-7, MK-8, MK-9, and MK-10) were assessed in eight different cheese products differing in ripening time, starter culture, fat content, and water content. The bioaccessibility was assessed using the static in vitro digestion model INFOGEST 2.0. Variation of the vitamin K content (<0.5 μg/100 g–32 μg/100 g) and of the vitamin K bioaccessibility (6.4–80%) was observed. A longer ripening time did not necessarily result in an increase of MKs. These results indicate that the vitamin K content and bioaccessibility differs significantly between different cheese products, and the ripening time, starter culture, fat content, and water content cannot explain this difference.

## 1. Introduction

Vitamin K was first discovered by the Danish scientist Henrik Dam as a coagulation factor in chicken [1]. It is now known that vitamin K describes a group of fat-soluble vitamers that all contain a 2-methyl-1,4-naphthoquinone ring structure with an isoprenyl side chain situated at position 3 varying in length and saturation (Figure 1) [2]. Vitamin K can be divided into two groups: phylloquinone (PK) (Figure 1A) and menaquinones (MKs) (Figure 1B).

Phylloquinone is produced by plants and phototropic cyanobacteria where it functions as an electron transporter in photosystem I, whereas menaquinones are produced by animals (MK-4), archaea, and bacteria (long chained MKs) where they are a part of the anaerobic respiratory and photosynthetic electron transport chain in the membrane [2,3]. MKs are most commonly found as MK-4 to MK-10, but MK-1 to MK-14 have been discovered in nature [4]. Vitamin K vitamers are co-factors that activate the vitamin K-dependent ɣ-glutamyl carboxylase that converts specific peptide-bound glutamate (Glu) residues into γ-carboxyglutamate (Gla) residues which are, hereby, activated and now can bind calcium [5,6]. These proteins are also referred to as Gla-proteins. Gla proteins are not only proteins that are associated with coagulation but also with bone and cardiovascular mineralisation, diabetes, cancer, and immune response [5,7]. The same associations are not found for phylloquinone [8,9,10]. These differences in response to phylloquinone and menaquinones intake have not been fully elucidated and need further study.

To date, only an adequate intake of PK is set (70 µg/day for adults), and this is only based on the coagulation ability of phylloquinone [11,12]. It is stated by EFSA that the knowledge of MKs in food and the bioavailability of these are too limited for EFSA to set a daily reference intake [12].

Vitamin K, together with other highly lipophilic compounds (logP > 8), is thought to be incorporated into mixed micelles in the small intestinal so as they can be transported over the mucus layer and are available for absorption by the epithelial cells in the small intestine [13,14,15,16].

The main dietary sources for PK are green vegetables, whereas the main source of MK-4 is meat and other animal products. Long chained MKs are found in fermented food, such as cheese, sauerkraut, and natto [17,18,19,20,21,22]. However, the content of different types of cheese is varying significantly from product to product and between different production locations [17,18,19].

The bioavailability of a compound depends on the bioaccessibility, absorption, distribution, metabolism, and excretion [15]. The bioaccessibility is typically the rate-limiting effect of the bioavailability for highly lipophilic compounds, such as vitamin K [15], and is, therefore, of outmost importance. The bioaccessibility of phylloquinone from vegetables is thought to be low, whereas the bioaccessibility of animal products and fermented products is higher. Because of the amount of cheese present in the diet, it is thought that cheese contributes to a significant amount of MKs in the diet [17,23]. Thus, the focus of this study was to investigate the bioaccessibility of vitamin K in cheese in order to investigate if difference in bioaccessibility should be taken into account when estimating, e.g., dietary intake based on the content of the vitamin K in the food product and the relation to health effects.

The aim of this study was to quantify the content of eight vitamin K vitamers (PK, MK-4, MK-5, MK-6, MK-7, MK-8, MK-9, and MK-10) in eight different cheese products and to assess if there is a difference between batches of the same cheese product. We further aimed to assess the bioaccessibility of the eight vitamin K vitamers in the five different cheese products using the static in vitro digestion model INFOGEST 2.0 [24].

## 2. Materials and Methods

### 2.1. Samples

Representative samples of eight cheese samples were send from Arla Foods amba, Viby J, Denmark (Arla) to the Technical University of Denmark, Kgs. Lyngby, Denmark. Specification on the shipped amount of cheese is described in Table A1. The cheese product names were as follows: Castello Danablu Extra Creamy with ripening time of 3 months (Danablu 3 M), Castello Danablu Extra Creamy with ripening time of 8 months (Danablu 8 M), Mild Cheddar Cheese with a ripening time of 3 months (Cheddar 3 M), Mature Cheddar Cheese with a ripening time of 12 months (Cheddar 12 M), a hard cheese of the Danbo type (Tistrup) with ripening time of 40 weeks (Tistrup 40 weeks), Tistrup with a ripening time of 10 weeks (Tistrup 10 weeks), Castello Brie Extra Creamy (Brie), and Hirtenkäse (Hirtenkäse). Information about the ripening time, starter culture, fat, and water content of the cheese products were received from Arla. A detailed overview of the samples is found in Table 1. Each sample was cut into approximately 1 cm^3^ cubes, frozen in liquid nitrogen, and homogenised in a coffee grinder (EGK 200, Rommelsbacher, Germany) for 20 s. The frozen cheese powder was mixed separately in a 5 L glass beaker, divided into 50 mL amber plastic containers, and stored at −20 °C until analysis.

### 2.2. Determination of Water in the Cheese Products

The water content of each cheese product was determined, by weighing an alufoil container wherein 5 g of cheese was added. The sample was then heated in an incubator at 102 °C for 18 h. The cheese sample was hereafter weighed, and the percentage loss of weight was defined as the water content of the cheese type. The water content was assessed in triplicates.

### 2.3. In Vitro Digestion of Cheese Products

The digestion of the cheese samples were done using the static in vitro digestion model INFOGEST 2.0 according to the published procedure [24]. In brief, the enzyme activity was assessed, solutions prepared, and pH tests of samples performed. The first phase of the in vitro digestion system is the oral phase, where 1 g of cheese was weighed into a 15-mL amber Eppendorf tube, and 1 mL of simulated salivary fluid was added. The sample was hereafter incubated for 37 °C for 2 min in an overhead rotator. The gastric phase began now by addition of 2 mL of simulated gastric fluid, together with gastric lipase (60 U/mL) and pepsin (2000 U/mL), and the pH was adjusted to 3.0. The sample was then incubated at 37 °C for 2 h in an overhead shaker. In the intestinal phase, 4 mL of simulated intestinal fluid was added to the sample, together with pancreatin (trypsin activity of 100 U/mL) and bile (10 mM bile salt), and the pH was adjusted to 7.0. Followed by incubation at 37 °C for 2 h in an overhead shaker. AT last, centrifugation for at least 30 min at 37 °C at 5500× *g* was performed until the solid phase was separated from the micellar phase. The micellar phase was transferred to another amber 15-mL Eppendorf tube, and the weight of the micellar phase was recorded. The micellar phase was now ready for analysis of vitamin K vitamers by the LC-ESI-MS/MS method described in Section 2.4. Bioaccessibility of each cheese product was assessed in triplicate.

### 2.4. Extraction and Quantification of Vitamin K Vitamers

The extraction and quantification was performed according to a previously published method [25], with few modifications, as described in Appendix B. In brief, 0.5 g samples (either sample of cheese or the micelle phase from the digestion model) was weighed in, and 125 ng of internals standards (IS) were added (deuterium labelled (d) PK (d7-PK), d7-MK-4, d7-MK-7, and d7-MK-9 (IsoScience LLC, Ambler, PA, USA)). Vitamin K and the IS was extracted from the food matrix using 2-propanol, n-heptane, and water. The fat was then removed from the sample using lipases, followed by a silica solid phase extraction clean-up step. The sample was dissolved in methanol and quantified using LC-ESI-MS/MS, as described elsewhere [26]. The LC-ESI-MS/MS MRM parameters are described in Table A2.

### 2.5. Statistical Analysis

LC-ESI-MS/MS instrument control and acquisition was done using Mass Hunter Workstation software (LC/MD data Acquisition version B08.00, Quantitative Analysis version B0701, Qualitative Analysis version B07.00, Agilent Technologies, Santa Clara, CA, USA).

The bioaccessibility (*B_acc_*) of vitamin K vitamers in the different cheese were calculated using the following equation:(1)$$B_{acc}\left(\%\right)=\frac{m_{M}}{m_{I}}*100$$
where *m_M_* describes the content of the vitamin K vitamer in the micelle phase at the end of the intestinal phase of the INFOGEST 2.0 model, and *m_I_* describes the total amount of the vitamin K vitamer in the cheese used for the INFOGEST 2.0 model.

One-way ANOVA tests was used to test for significant (*p*-value < 0.05) differences between batches of the same cheese product, and linear regression was used to tests if the ripening time, fat, or water content significantly (*p*-value < 0.05) affected the content of vitamin K vitamers and the bioaccessibility of vitamin K vitamers.

The PK equivalence (PKeq) was calculated based on the differences in molecular weight of the vitamers to express the total amount of vitamin K in the cheeses.

All statistical work was performed using RStudio (© 2009–2019 RStudio, Inc., Version 1.2.1335, Northern Ave, MA, USA) and Excel (Microsoft^®^ Excel^®^ 2016 (16.0.5017.1000), Microsoft, Redmond, WA, USA). Results are in general given as mean ± standard deviation (SD).

## 3. Results

### 3.1. Content of Vitamin K in Cheese Products

In this study, eight cheese products were studied. A detailed overview of the different cheese types with ripening time, starter culture, fat content, and water content is shown in Table 1.

The content of vitamin K vitamers in the different cheese products and ripening times were quantified and are shown in Figure 2A. The content of PK ranged from 16 ng/g cheese to 38 ng/g cheese, the content of MK-4 ranged from 57 ng/g cheese to 83 ng/g cheese, the MK-5 content ranged from <5 ng/g to 25 ng/g, no MK-6 was found in any of the samples, the content of MK-7 ranged from <5 ng/g to 32 ng/g, the content of MK-8 ranged from <10 ng/g to 169 ng/g, the MK-9 content ranged from <10 ng/g to 323 ng/g, and no MK-10 was found in any of the samples (Table A3). The total vitamin K content of the cheeses ranged from 153 ng PKeq/g to 476 ng PKeq/g. A serving of cheese (1 slice of approximately 28 g) could, therefore, contribute 4.3 µg to 13 µg, corresponding to 6% to 19% of the adequate intake set for PK. These vitamin K vitamer contents were comparable with what has been found by others; however, there is a great variation between and within cheese types produced in different places in Europe, [17,19,27,28]. Remarkably higher content of vitamin K vitamers were found in a study of cheese from the U.S., indicating that the production location may significantly affect the vitamin K [18].

It was tested whether the content of the vitamin K vitamers correlated with the fat and water content in the cheeses using linear regression. Based on the statistical analysis (Section 2.5), it was concluded that the starter culture, fat, and water content cannot alone explain the vitamin K content in the cheese.

### 3.2. Development of Vitamin K Content during Ripening of Cheese

It was further tested whether the content of vitamin K was dependent on the ripening time of the cheese. Three cheese products (Tistrup, Danablu, and Cheddar) were analysed at different ripening times. For Tistrup with a ripening time of 10 weeks and 40 weeks, it was observed that the content of PK, MK-8, and MK-9 increased significantly from 10 to 40 weeks of ripening time (Figure 3A). For the Danablu cheese, it was observed that the content of MK-4, MK-7, and MK-9 increased significantly from 3 to 8 months of ripening time (Figure 3B). In the cheddar cheese, a significant decrease in content of MK-5, MK-7, and MK-9 was observed from 3 to 12 months of ripening (Figure 3C).

The content of PK and MK-4 are assumed to originate from the milk from which the cheese is produced and not from the bacteria in the cheese [29]. The significant difference in PK and MK-4 content observed in Danablu and Tistrup cheese may be due to loss of water that is observed between the two ripening times of 5 months and 30 weeks, respectively (Table 1). The milk used for the production of cheeses may influence the content of PK and MK-4 of the cheese, as seasonal changes has been observed for the PK content in milk [30,31]. However, the influence of the origin of milk on the produced cheese was not investigated in this study.

The Danablu and Tistrup cheeses were made based on the same starter culture, whereas the cheddar cheese were made with a different starter culture (Table 1). Others have observed that different starter cultures result in a difference in content of vitamin K vitamers [28]. It has further been observed that the content of vitamin K in cheese may decrease for some vitamers and increase for others dependent on the bacteria species present in the cheese [28]. The ability for production of different vitamin K vitamers differs between bacteria species and even between strains [19,28,32]. Others have found that the ripening time of the Norwegian Gamalost does not significantly affect the vitamin K content [32]. The vitamin K content seems to depend not only on the ripening time, but more studies are needed to fully understand the processes taken place during ripening of cheese.

### 3.3. Differences between Different Batches

It was tested whether there was a significant difference in the content of vitamin K vitamers between different batches of the same cheese product. For this, the salad cheese Hirtenkäse was chosen, and four batches produced successively in the last quarter of 2020 were analysed for vitamin K (Figure 4). No significant difference in vitamin K content was observed between any of the batches, indicating that the production is standardised to a degree that does not cause variation in the content of vitamin K. Further studies are needed to insure that this is the case in general for production of cheese in Denmark, and health declarations might be of interest [33]. A great variation within the same cheese products produced at different dairies has been found; however, it is not clear whether these differences were seen, as well, between different batches from the same cheese product [19]. As the production procedure of the Hirtenkäse cheese is not thought to change between the different bathes, it was hypothesised that no significant difference in content of vitamin K would be found between the batches, which was what was observed (Figure 4).

### 3.4. Bioaccessibility of Vitamin K in Cheese

To assess the bioaccessibility in vivo can be demanding and ethically challenging, and an in vitro digestion model is, therefore, of preference when screening of many food matrices are of interest [24,34,35,36]. Others have shown that, for carotenoids and vitamin E, the in vitro determined bioaccessibilities are comparable with in vivo determined bioaccessibilities [13].

In this study, the bioaccessibility of the different cheeses was determined (Figure 2). When looking at the amount of PK that becomes bioaccessible when digesting 1 g of cheese, the amount from Danablu 3 M and 8 M and Hirtenkäse is higher than for Brie and Cheddar 3 M and 12 M, which, again, is higher than for Tistrup 10 weeks and 40 weeks (Figure 2). The same pattern is observed for the bioaccessibility of PK and MK-4, where the highest bioaccessibilities are found in Danablu 8 M (53% for PK and 80% for MK-4), and the lowest is found in Tistrup 40 weeks (6.4% for PK and 7.7% for MK-4).

Danablu 8 M was the only product that had a quantifiable amount of MK-7 after digestion, where a bioaccessibility of 58% was observed. Bioaccessibility of MK-8 was estimated in Danablu 8 M and 3 M and in Tistrup 40 weeks with bioaccessibilities of 46%, 28%, and 17%, respectively. These corresponded with observed bioaccessibility of PK and MK-4.

For MK-9, the bioaccessibility was quantified in all but the Brie and the Cheddar 3 M. For the rest of the chesses, the order from highest to lowest assessed bioaccessibility followed the pattern observed for PK and MK-4, with Danablu 8 M having the highest bioaccessibility of MK-9 (58%) and the lowest in Tistrup 40 Weeks (8.5%).

The total amount of bioaccessible vitamin K was highest in Danablu 8 M, which was significantly higher than total amount of vitamin K bioaccessible in Danablu 3 M, which, again, was significantly higher than the rest of the cheese products.

The Danablu cheese products are harbouring the highest total content of vitamin K, with Tistrup 40 weeks having the next highest content. Even though a relatively high content of vitamin K is found in Tistrup 40 weeks, it has a very low bioaccessibility compared with the other types of cheese. This indicates that the bioaccessibility is not dependent on the content of vitamin K in the cheese. Neither the fat content nor the moisture of the cheese or the different starter cultures can explain the bioaccessibility found in the different cheeses based on the statistical analysis (Section 2.5). However, it is a limitation that the fat content was not analysed in the products but, as mentioned, received from Arla. The decrease in water content indicates an increase in fat content during ripening.

Studies into the bacteria *Propionibacterium freudenreichii*, which is commonly used in production of fermented dairy products, showed that, in the growth period of *P. freudenreichii*, menaquinones were accumulated inside the cells, but, when the bacteria cells started lysing after depletion of lactose, the menaquinones was released from the cells [37]. It could be that, in Danablu 8 M, more cells have lysed and released the content of menaquinones, making them more bioaccessible. This theory correlates with a higher bioaccessibility in Danablu 8 M compared to Danablu 3 M. If the lysing of the bacteria cells have not happened to the same degree in the Tistrup cheeses, the MKs would maybe not be as bioaccessible as in Danablu. However, this cannot explain the higher bioaccessibility of PK and MK-4 in Danablu compared to other cheese products as it is hypothesised that these are originating from the milk and not the bacteria [29].

To our knowledge, to date, no assessment of bioaccessibility of vitamin K has been reported. Our results show that bioaccessibility varies between the vitamers, as well as between cheeses; thus, information on relative bioaccessibility could be taken into account when calculating dietary intake. This will allow a more accurate assessment of dietary intake of vitamin K, which may improve the correlation to physiological response. Furthermore, any difference between the MKs and PK is relevant when establishing recommended daily intake of MKs and, potentially, updates for the PK value. More studies are needed to investigate the factors affecting the vitamin K bioaccessibility in cheese.

The bioaccessibility is, in general, the rate limiting factor for the bioavailability for fat-soluble compounds [15]. However, others have found that, in some cases, the food matrix is limiting the bioaccessibility of a compound, while simultaneously increasing the absorption of the same compound [37], so, extrapolations of the effect on of the bioaccessibility for the bioavailability should be done with caution.

## 4. Conclusions

We showed that the vitamin K vitamer content varied considerably within Danish cheese products. The highest bioaccessibility was found in Danablu with a ripening time of 8 months (53% for PK, 80% for MK-4, 58% for MK-7, 46% for MK-8, and k8% for MK-9) and the lowest bioaccessibility was found in Tistrup with a ripening time of 40 weeks (6.4% for PK, 7.7% for MK-4, 17% for MK-8, and 8.5% for MK-9). The content of vitamin K, the starter culture, fat, or water content could not explain the differences in bioaccessibility of vitamin K in the different cheese products. Based on these data, it seems that some cheeses may be good sources of vitamin K, while the low bioaccessibility of other cheeses limits their importance as a good source of vitamin K. These differences in relative bioaccessibility are important to take into consideration when relating physiological condition to the vitamin K intake. However, more studies are needed to further elucidate the factors determining the bioaccessibility of cheese and whether the bioaccessibilities of these limits their bioavailability.

## Figures and Tables

**Figure 1 foods-10-02938-f001:**
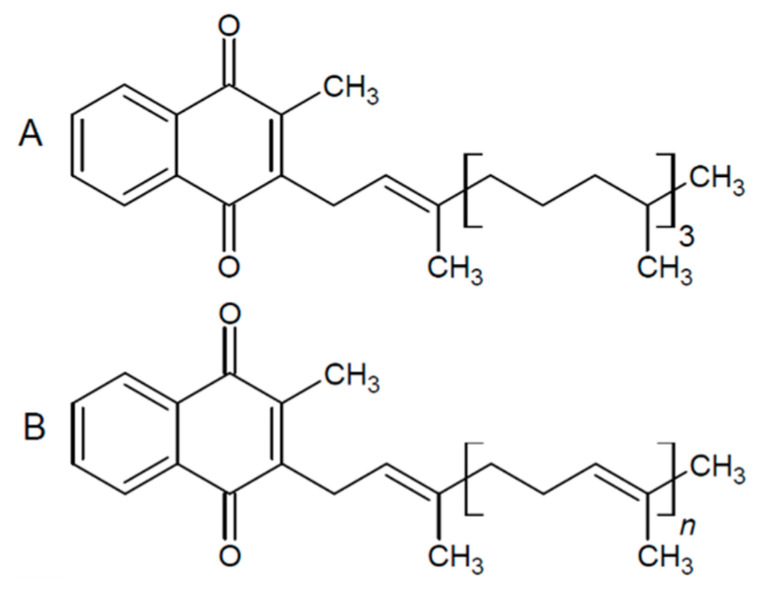
Structure of phylloquinone (PK) (**A**) and menaquinones (MKs) (**B**).

**Figure 2 foods-10-02938-f002:**
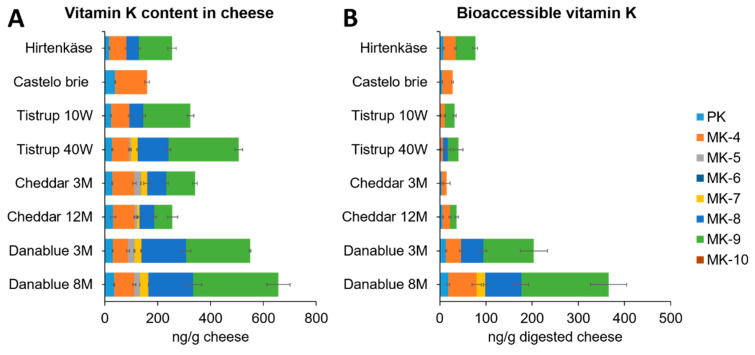
Vitamin K content (ng/g cheese) in the different cheese products (**A**) and the bioaccessible fraction of vitamin K in 1 g of digested cheese (**B**). Error bars depicts the standard deviation. Each sample was analysed in triplicates (*n* = 3).

**Figure 3 foods-10-02938-f003:**
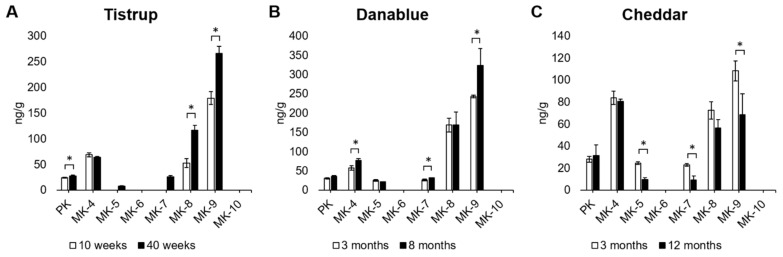
Vitamin K vitamer content in three different cheese products at different ripening times. (**A**): Tistrup cheese, 10 and 40 weeks ripening. (**B**): Danablu cheese, 3 and 8 months ripening. (**C**): Cheddar cheese, 3 and 12 months ripening. Error bars depicts SD for the three replicates. *: indicate significant difference between the two ripening times for the same vitamin K vitamer.

**Figure 4 foods-10-02938-f004:**
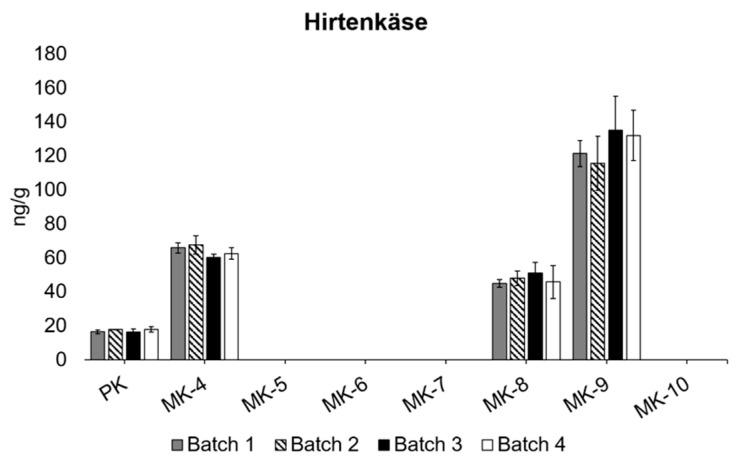
Four batches of Hirtenkäse analysed for content vitamin K vitamers (ng/g). All samples were analysed in triplicates (*n* = 3). Error bars depicts SD.

**Table 1 foods-10-02938-t001:** Cheese cultures with ripening time, starter culture, fat content, and water content.

Cheese Product	Ripening Time ^1^	Starter Culture ^1^	Fat (%) ^1^	Water (%) ^2^
Danablu	3 months	*Lactococcus lactis lactis,* *Lactococcus lactis cremoris,* *Lactococcus lactis lactis diacetylactis,* *Leuconostock mesenteroides cremoris*	37	41
	8 months			38
Tistrup	10 weeks		26	44
	40 weeks			39
Brie	3 weeks		39	42
Hirtenkäse	1 week		28.6	57
Cheddar	3 months	*Lactococcus lactis lactis,* *Lactococcus lactis cremoris*	35	36
	12 months			37

^1^ Information from Arla. ^2^ Analysed in triplicate.

## Data Availability

The data used for this study are available from the corresponding author upon request.

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
