# Peer review of "Content and Bioaccessibility of Vitamin K (Phylloquinone and Menaquinones) in Cheese"

_foods, 2021, doi:10.3390/foods10122938_

Round 1

Reviewer 1 Report

This work presents interesting data on vitamin K contents and bioaccessibility in cheese, the latter being a notable novel contribution to the literature. Methods are adequate and results are properly discussed, although some corrections should be done in the text to improve clarity.

Introduction

Line 35. Please check comma and “and” use: menaquinones are produced by animals, archaea and bacteria, ...

Line 42. A verb is missing in the sentence: Gla proteins are not only proteins that (are) associated with coagulation

Line 49. The sentence is not properly written, do you mean the knowledge is too limited to set a daily reference intake?

I recommend revision of the English language throughout the manuscript: “the main dietary sources for PK are” instead of “the main dietary sources for PK is” (line 55), “whereas” instead of “where as” (line 55), “from product to product” instead of “from product, to product” (line 58), “therefore” instead of “there for” (line 63), “and” instead of “end” (line 199)...

Materials and methods

Extraction and quantification of vitamin K vitamers. The method is described in Appendix B, not appendix A (line 119). Please describe the supplier of the deuterated standards. Was the same procedure applied to micellar phases from digested samples?

Table 1. Scientific names should be written in italics.

Table 1. Fat contents are apparently wrong, as no change is shown over ripening time. Given that water content decreased, logically the percentage of fat should increase.

Results

Differences between different batches. Further information on the batches, if available, would be helpful to interpret the results, such as time difference between batches production, factory (if different factories produced the cheese), season (different season might induce differences in milk properties)...

Bioaccessibility of vitamin K in cheese. Lines 228-230, a verb is missing, the sentence should be rewritten for clarity.

Bacteria scientific names should be written in italics (lines 253, 254, etc.).

Appendix B

Some text referring to instructions to the authors remained at the end of the appendix, it should be removed.

Reviewer 2 Report

The manuscript submitted to Foods by Jenssen et al., titled: "Content and bioaccessibility of vitamin K (phylloquinone and menaquinones) in cheese" is aiming to assess the presence and bioavailability of vit K in cheeses. The topic is somewhat interesting from a food science perspective with potential nutritional implications. The manuscript is well-written with good flow and reasonable structure. The language is appropriate and it is well balanced although somewhat lacking in references.

The reviewer would like to offer the following points for consideration:

  1. The authors use the term bioaccessibility and assess that via in vitro assays. In the reviewer's opinion what is meaningful to know is the bioavailability which alludes to the amount of compound that becomes available and biointegratable in a biological system post digestion/absorption.
  2. Conceptually it is difficult to see the novelty in the work. While it is interesting to know how much vitamin K activity compounds exist in different cheeses it is difficult to see what advances in science and in the field that knowledge entails. 
  3. The authors state that there were differences in the content of vitK compounds depending on the fermentation time and similar parameters. This is reasonable and normal to expect given the mechanism via which vitamin K is produced. So again it is challenging to view the novelty here.

Reviewer 3 Report

good information and it helps to update the literature about vit K in dairy foods.

Pleas e review other recent work related to the presence of Vit K in dairy and reason for differences.   could be milk source.. feeding system. processing steps.  remeber the vitamin is asscoiated with fat members, it could be altered during processing of milk.... some milk is homogenized before cheese making.  etc......I wish you add more in the discussion.  starter culture usually does not do much with vit K.  check work done in Noway in the 80's food science.  

Round 2

Reviewer 2 Report

the authors have made a reasonable effort in addressing reviewer’s comments.

Author Response

Thank you